# Assessing Differences in the Implementation of Smoke-Free Contracts—A Cross-Sectional Analysis from the School Randomized Controlled Trial X:IT

**DOI:** 10.3390/ijerph18042163

**Published:** 2021-02-23

**Authors:** Lotus Sofie Bast, Susan Andersen, Stine Glenstrup, Mogens Trab Damsgaard, Anette Andersen

**Affiliations:** 1National Institute of Public Health, University of Southern Denmark, Studiestraede 6a, 1455 Copenhagen, Denmark; sua@niph.dk (S.A.); sgla@sdu.dk (S.G.); trab@niph.dk (M.T.D.); 2Steno Diabetes Center, Hedeager 3, 8200 Aarhus, Denmark; aneta2@rm.dk

**Keywords:** school-based smoking prevention, implementation, adolescents, social inequality, gender differences

## Abstract

Objective: The X:IT study is a school-based smoking preventive intervention that has previously been evaluated in a large randomized controlled trial (RCT) with good effects. However, the actual effect for participating students depends on the individual implementation. The aim of this study was to examine the implementation of smoke-free contract, which is one of the three main intervention components. Specifically, we examined whether it was implemented equally across family occupational social class (OSC), separately for boys and girls, the joint effect of OSC and gender, and the participants’ own reasons for not signing a contract. Results: Overall, the smoke-free contract was well implemented; 81.8% of pupils (total N = 2.015) signed a contract (girls 85.1, boys 78.6%). We found a social gradient among girls; more than 90% were in OSC group I vs. 75% in group VI. Among boys, however, we found no difference across OSC. Boys in all the OSC groups had about half the odds (i.e., medium OSC boys: OR = 0.48 (95% CI: 0.32–0.72) of having a smoke-free contract compared to girls from a high OSC. Conclusion: future interventions should include initiatives to involve families from all OSC groups and allow for different preferences among boys and girls.

## 1. Introduction

School-based smoking preventive initiatives are widely used, and, if well implemented, have the possibility to help prevent a new generation from starting to smoke [1]. They may further help prevent or decrease socioeconomic inequality later in life [2].

Some have argued that interventions themselves may generate or increase socioeconomic inequalities in smoking prevalence [3]. This may be due to different aspects of the implementation process—i.e., access to materials and tools, uptake, and compliance [4]. Therefore, it is important that smoking preventive initiatives aim at being equally effective across socioeconomic groups [5] or at being more effective among individuals from low socioeconomic positions.

Generally, the best effect of school-based smoking prevention is seen in interventions that target adolescent smoking from multiple sides simultaneously—i.e., structural as well as individual initiatives [1]. Among structural initiatives are, for example, smoke-free policies implemented for all pupils and/or staff at the same time, whereas smoke-free contracts or dialogues between parents and children are oriented at the individual pupil and require action from each individual participant. Therefore, there is a risk that especially individually oriented initiatives in the school setting may be implemented differently among pupils, causing inequalities in the intervention effect [6].

In Denmark, there is no significant difference in smoking among boys and girls in younger age groups [7]. However, boys and girls may have different reasons for smoking initiation [8], and gender may also play an important role in the implementation of smoking preventive activities. Among other factors, gender affects decision-making, engagement, and preferences for intervention uptake [9]. Overall, there is a variety of determinants for adolescent smoking at different levels—i.e., in individual, social, and societal levels. Adolescents with lower socio-economic backgrounds are at higher risk of smoking initiation, as well as adolescents with low academic performance [10,11]. At the social level are peer and parental behavior and attitudes, whereas the societal level includes restrictions and advertising [11].

Most evaluations of school-based smoking prevention have not addressed these aspects and therefore the mechanisms behind any inequality in individual intervention uptake is unknown. We hypothesized that families of higher socioeconomic position more often would sign the smoke-free contract. To our knowledge, there are no studies of the implementation of smoke-free contracts in relation to family socioeconomic position or gender. Therefore, the aim of this study was to examine the implementation of a smoke-free contract. Specifically, we examined the intervention uptake across family occupational social class (OSC) separately for boys and girls, the joint effect of OSC and gender, and the participants’ own reasons for not signing a contract.

## 2. Methods

### 2.1. Evaluation of the X:IT Intervention

This study is based upon data from an effective smoking preventive intervention X:IT which was evaluated in a cluster randomized controlled trial with 97 schools in 2010–2013. Overall, X:IT was effective in preventing adolescent smoking; the odds ratio for smoking among pupils in intervention schools compared to the control group after one year of intervention was 0.61 (95% CI: 0.45–0.81) [12]. A thorough assessment of the school-wise implementation showed that one fourth of the participating schools succeeded in implementing all three intervention components as prescribed in the program. Schools with a high implementation showed reduced risk for pupil smoking, with an odds ratio of 0.44 (95% CI: 0.32–0.68) [13].

### 2.2. Study Population

All the 98 Danish municipalities were invited to join the X:IT study and 17 agreed to participate (17.3%). A total of 97 of 302 eligible schools were enrolled (32.1%). Schools were randomly assigned to either the intervention or control group using a stratified simple randomization procedure, meaning that, within each municipality, schools had an equal chance of becoming an intervention or control school. All the pupils from seventh grade (13-year-olds) were invited to participate (*n* = 4468). This paper was restricted to pupils from intervention schools with responses to baseline and first follow-up (*n* = 2015). See the flowchart in Figure 1.

### 2.3. The X:IT Intervention

The main intervention components were (1) smoke-free school grounds implemented at the school level (no smoking for pupils, teachers, or visitors at school), (2) curricular activities implemented at the school class level (at least eight lessons a year for three years from a specially developed curricular material), and (3) smoke-free contracts between parents and their children implemented at the individual level. Further details about the intervention can be found in the X:IT study protocol [14].

The smoke-free contract.

The smoke-free contracts were handed out to pupils at the beginning of the school year. Teachers handled the distribution of contracts during a school lesson. One part of the contract should be kept at home, preferably at a visible place, and the other part returned to the teacher. By signing the contract, the pupil promised to stay smoke-free for the following year, and the parent committed oneself to help create a smoke-free environment around the child.

### 2.4. Measures

Pupils answered a web-based questionnaire during a school lesson after a standardized instruction given by the teacher.

The outcome was the smoke-free contract. Pupils were asked to report whether they had signed a smoke-free contract in the beginning of the school year. We dichotomized responses into yes vs. no/I do not know.

Socioeconomic position was measured as family occupational social class (OSC) based on pupils’ responses to two items on father’s and mother’s occupations (work place and job function). The pupils’ information on parental occupation was coded from (I) high to (V) low social class and (VI) indicating parents receiving social benefits. Family occupational social class was then determined by the highest-ranking parent according to the standards of the Danish National Institute of Social Research. This coding is almost similar to the Registrar General classification used in the UK [15]. In the joint effect analysis, OSC was grouped into three categories: high (I+II), medium (III+IV), and low (V+VI).

Current smoking among pupils was measured by: How often do you smoke? Responses were dichotomized in “Daily”, “Weekly”, “Monthly”, or “More seldom” vs. “Never”.

Pupils, who responded ‘no’ to having a contract were asked about the reason: “Why didn´t you sign a contract?” Response options included: “I smoke”, “My parents wouldn´t sign the contract”, “I just didn´t do it”, or “I never received a contract”.

### 2.5. Statistical Analyses

Due to a substantial number of missing on the variable measuring occupational social class, we used a dataset with imputed values for the missing items. We performed multistage imputation and two rounds of 20 imputations as recommended by Graham (2012). [16]. The agreement between the original and the imputed data set was good [10].

We performed multiple logistic regression analyses to estimate the odds ratio for having a signed contract, with girls from a high OSC as the reference group. We used the Proc mianalyze procedure in SAS version 9.3. We provided a crude analysis and an adjusted analysis that considered pupils own smoking at baseline, as this may influence the probability of signing a smoke-free contract.

## 3. Results

Descriptive statistics are provided in the left part of Table 1. Overall, 81.8% of the pupils signed a contract. The proportion was higher among girls compared to boys—85.1 vs. 78.6% (*p* = 0.001).

As seen in Figure 2, there was a clear social gradient in contracts among girls, with more than 90% in OSC group I vs. 75% in group VI. Among boys, however, we found no differences across the socioeconomic groups.

When examining the combined effect of OSC and gender, we saw that the boys in all socioeconomic groups had about half the odds of signing a smoke-free contract compared to girls from high OSC. Girls in low OSC had odds that corresponded to boys—0.56 (0.33–0.97)—whereas the odds for girls from a medium OSC were 0.76 (0.50–1.19) (results presented on the right side of Table 1).

Among pupils with no smoke-free contract (49 girls and 65 boys), the main reasons for not having a contract were the same for boys and girls; that they never received a contract or that there were no specific reasons for not signing.

The question, “Why didn´t you sign a contract?”, was answered by “I smoke” (8.2% girls, 6.2% boys); “My parents wouldn´t sign the contract” (2.0% girls, 10.8% boys); “I just didn´t do it” (53.1% girls, 52.3% boys); “I never received a contract” (49.0% girls, 44.6% boys).

## 4. Discussion

We found that more girls compared to boys signed a smoke-free contract with their parents; however, the overall compliance for both genders was high. Further, we showed a social gradient in signed contracts for girls, so that more girls from higher OSCs had a contract. When examining the combined effect of OSC and gender, we saw that the boys from all OSC had about half the odds of signing a smoke-free contract compared to girls from the highest OSC. Only few studies examined the combined effect of OSC and gender among adolescents [17]. It has been argued that differences in socializing patterns between boys and girls may explain differences in health-related behavior according to socioeconomic position [17], which also may apply here. In example, boys tend to cope with development in a more externalized way, whereas girls tend to be more internally focused [17].

We found that girls from a higher OSC more often signed a contract, whereas boys in all OSCs and girls from the lower OSCs had lower proportions of signed contracts. Several studies have found a link between smoking and OSC [18,19]. A possible explanation for this is that parents with higher versus lower educational attainment may be more concerned about the health risks of smoking because of their greater knowledge about them [19]; however, it is surprising that our study did not detect a OSC gradient for boys. When trying to explain these OSC and gender-related differences, it can be important to look at the different steps in relation to the implementation of the contracts. First, contracts should be handed out to all participating pupils. As contracts were handed out by teachers during a school lesson, we assume no systematic differences according to socioeconomic position or gender in this step

Next, the contract was brought home by the children and (hopefully) shown to parents. Fewer boys than girls signed a contract, which may be related to this step; girls may generally be better at involving parents in school activities. However, the difference may also be explained by other factors such as maturity; boys in this age are more immature than girls—and boys may not perceive the contract as important, or the simply may forget the contract in the bag. However, we found no markedly different reasons for not having a contract when asking the children themselves.

Afterwards, the children and parents should preferably talk about the contract and smoking related issues in general and thereafter both sign the contract. We do not know how this is done and what happens at home in this situation. Two previous studies observed parents and children having conversations about smoking and tobacco and found positive associations between the quality of these conversations and not starting to smoke or not increasing smoke. None of these studies, however, examined whether there were differences according to gender of the child [20,21].

As a part of the evaluation of X:IT, we conducted interviews with parents from lower socioeconomic positions. Generally, parents from low socioeconomic positions did not recognize themselves and their children in the materials; the persons pictured did not look like them and the wording was too academic. This may explain some of the tendency towards the lower proportions of contracts among these groups, at least among girls. The missing social gradient among boys may imply different mechanisms for boys and girls. Boys may be more strongly influenced by other factors—i.e., youth culture or general peer influence—or families with teenage boys may not talk about smoking and tobacco-related issues to the same extent as families with teenage girls.

## 5. Limitations

Data were self-reported and thereby the results could be biased by social desirability for having a signed contract, as well as being a non-smoker. However, self-reporting is regarded as a valuable method of gaining insight into pupils’ explicit attitudes, experiences and behaviors, and we know from previous studies that adolescents can answer items about smoking with good validity [22,23]. Further, answers about signed contracts from the questionnaire data corresponded well with copies of the actual contracts, and therefore we believe that they are able to answer this question. Lastly, we assume that asking adolescents themselves about their own behavior provides the best picture of their lives. The use of questionnaire data among pupils limits our ability to gain insight into the situation where the families talked about tobacco and signed the contract. We cannot eliminate the risk that the quality of this situation is associated to factors such as parental smoking status. However, previous research underlines the fact that parents that are smokers themselves can talk to their kids about anti-smoking issues in an effective way [24].

The schools included in this study were not randomly selected. Participation could be motivated by a high prevalence of smoking or a high level of resources and a general focus on health, potentially more profound among schools with low levels of smoking. The direction of this possible selection bias is unknown. However, the possibility for each participating school to become an intervention or a control school was completely random, and this issue is assumed to have no influence on the social gradient in signed smoke-free contracts.

## 6. Conclusions

From a public health point of view, the differential implementation of the contract according to socioeconomic position could potentially widen the social inequality in smoking. Future interventions should ensure that all OSC groups are equally represented in the selection and randomization and take into account the fact that different strategies may appeal to adolescents and families of different socioeconomic positions. Further, boys and girls may benefit from different initiatives.

## Figures and Tables

**Figure 1 ijerph-18-02163-f001:**
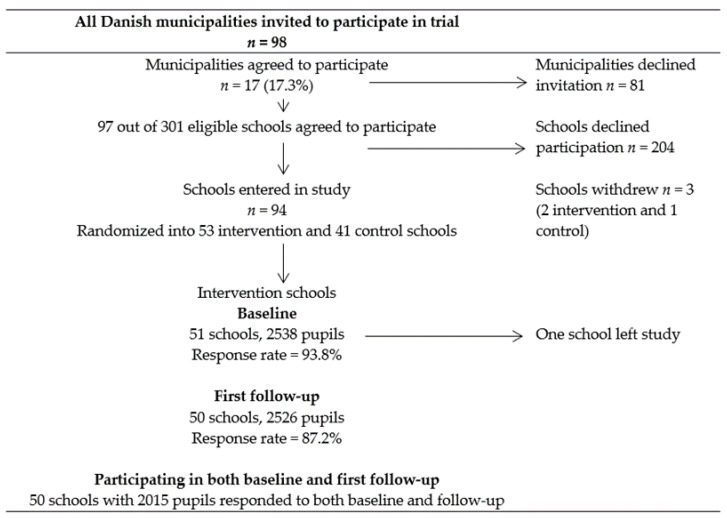
Flow diagram of recruitment, randomization, and participation in the X:IT study.

**Figure 2 ijerph-18-02163-f002:**
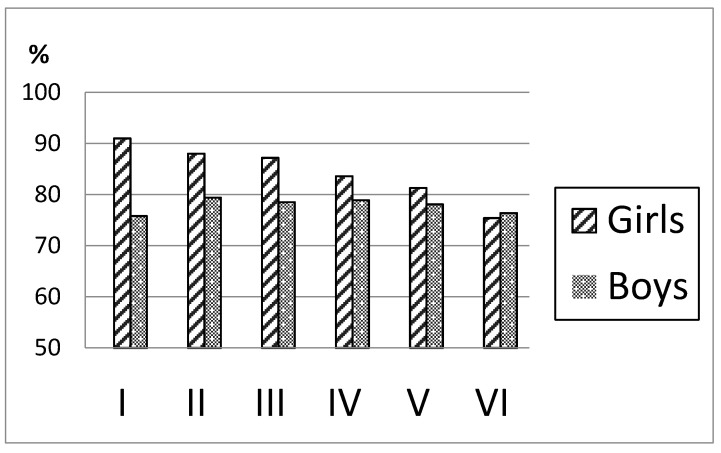
Proportion of girls and boys with a signed smoke-free contract in OSC I-VI.

**Table 1 ijerph-18-02163-t001:** Descriptive statistics and analyses of joined effects of gender and OSC on the odds ratio (95% CI) of having a smoke-free contract.

	Girls 49.8% (*n* = 1003)	Boys 50.2% (*n* = 1012)
**Smoke-free contracts**	85.1% (854)	78.6% (795)
**Occupational social class**		
I	5.1% (51)	5.9% (60)
II	29.6% (297)	26.5% (268)
III	14.8% (149)	18.7% (189)
IV	33.9% (339)	31.9% (323)
V	11.8% (118)	12.6% (128)
VI	4.9% (49)	4.4% (44)
**Current smokers at baseline**	4.6% (46)	5.1% (51)
	**Crude model**	**Adjusted model ***
**Occupational social class**		

Girls high	1 (reference)	1 (reference)
Girls medium	0.69 (0.47–1.11)	0.76 (0.50–1.19)
Girls low	0.51 (0.30–0.86)	0.56 (0.33–0.97)
Boys high	0.48 (0.31–0.75)	0.49 (0.32–0.77)
Boys medium	0.48 (0.32–0.72)	0.51 (0.34–0.77)
Boys low	0.45 (0.27–0.75)	0.49 (0.29–0.82)
**Current smoking**		
No	-	1 (reference)
Yes	-	0.25 (0.17–0.39)

* Adjusted for current smoking.

## Data Availability

Data are available upon request.

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
