# Peer review of "Assessing Differences in the Implementation of Smoke-Free Contracts—A Cross-Sectional Analysis from the School Randomized Controlled Trial X:IT"

_ijerph, 2021, doi:10.3390/ijerph18042163_

Round 1
Reviewer 1 Report
Lines 44-46.
The authors use the term "gender"; I think the correct term should be "sex" (PMID:27788671).
Lines 195-200
Previously, the authors describe variables of interest in their study population (PMID: 26210612, Table 1) and show the percentage of mother and father smokers. Being the same study population in this paper; How can this interfere with the conversation parents have with their children? (PMID: 18625061); and, how could you evaluate the “quality of this conversation”?
I consider that the study has a limitation because the results show the behavior of the boys and girls who attend a school. What happens to those who, because of an even lower social level, cannot attend school? Do you think that this unexplored population could behave like level VI of this paper? Or, would they need to expand the study to include this unmonitored population?
Author Response
Dear reviewer,
Thank you for conducting the review. Please find answers to your questions below.
Best Lotus Bast
Lines 44-46.
The authors use the term "gender"; I think the correct term should be "sex" (PMID:27788671).
RESPONSE: Thank you for this comment. We have given this a lot of thought; however, we prefer to keep the term “gender” instead of “sex”. We use the term throughout the whole manuscript.
This is from the perspective that sex is more about biological attributes, whereas gender goes on more social–cultural aspects. See i.e. Kalenga et al. 2020: Sex and gender considerations in health research: a trainee and allied research personnel perspective, or Clayton & Tannenbaum, 2016: Reporting Sex, Gender, or Both in Clinical Research?
Lines 195-200
Previously, the authors describe variables of interest in their study population (PMID: 26210612, Table 1) and show the percentage of mother and father smokers. Being the same study population in this paper; How can this interfere with the conversation parents have with their children? (PMID: 18625061); and, how could you evaluate the “quality of this conversation”?
RESPONSE: Parental smoking status may have an impact on the conversation, in fact we hypothesized that families of lower SEP would tend to have less smoke free agreements. As we have no data from the parents, we can’t evaluate the quality of the conversation. We have added this as a limitation for the study. See lines 227-232.
I consider that the study has a limitation because the results show the behavior of the boys and girls who attend a school. What happens to those who, because of an even lower social level, cannot attend school? Do you think that this unexplored population could behave like level VI of this paper? Or, would they need to expand the study to include this unmonitored population?
RESPONSE: This is a good question. However, in Denmark almost all children attend either a public or a private school. You are allowed to home school your child, but no child is left to their own without municipal contact. Those, who have difficulties attending a school in Denmark gets a lot of municipal support and are still enrolled in the normal school system. Hence, there is no such unmeasured population.
Reviewer 2 Report
This brief report provides helpful insights into gender and socioeconomic-related disparities in the adoption of a school-based smoking prevention intervention. Specifically, the study demonstrates a marked gradient in the signing of a smoke-free contract, based on family socioeconomic position (SEP), for girls but not boys. Boys ad an overall lower rate of completion of the smoke-free contract. The authors highlight potential divers of this effect and point to strategies to address this apparent source of disparities in adolescent smoking initiation.
Overall, the paper is well-written and the information is clearly communicated. The data are analyzed appropriately, and the conclusions are justified.
However, several issues are noted which require consideration.
There is inconsistent or incorrect use of terminology used in connection with the field of implementation science. While the authors correctly note that implementation is a process, entailing a set prescribed activities, it is misleading to refer to an intervention as “well-implemented” or “high implementation” without a more formal analysis of the process that informs that implementation. Instead, the authors appear to be referencing the “reach” of the intervention, which, in this study, encompassed the proportion of participants who “adopted” the intervention. Thus, it is strongly suggested that the manuscript is carefully edited to more appropriately employ the terms implementation, adoption, and reach.
Line 44: “reasons for smoking initiation” – could be more clearly expressed to distinguish between individual reasons/motives and drivers of initiation, including risk factors such as low SEP. I suspect that the latter is more relevant.
Line 118: A reference is needed to support the OSC metric used in this paper
Table 1 appears to have experienced formatting problems and is very hard to comprehend in its current form. The table is also confusing because multiple concepts are being communicated, including descriptive data and regression models. It is further confusing to see the overall proportion of girls vs. boys who signed a contract in the table, while underneath is the distribution of the sample by social class, rather than the actual proportion who signed the contract by social class (which appears in the figure). Revisions are needed to make the table more comprehensible, including the possibility of breaking the table into 2 smaller tables.
Lines 185-86: The authors claim that distribution of contracts was unbiased because they were systematically distributed through schools. However, this does not account for potential OSC variations between schools. The authors should provide more information about the socioeconomic positions of the schools that took part in the study, and explain whether a cluster analysis might have allowed better control of variations in OCS.
Author Response
Dear reviewer,
Thank you for conduction this review. Please find responses below.
Best Lotus Bast
This brief report provides helpful insights into gender and socioeconomic-related disparities in the adoption of a school-based smoking prevention intervention. Specifically, the study demonstrates a marked gradient in the signing of a smoke-free contract, based on family socioeconomic position (SEP), for girls but not boys. Boys ad an overall lower rate of completion of the smoke-free contract. The authors highlight potential divers of this effect and point to strategies to address this apparent source of disparities in adolescent smoking initiation.
Overall, the paper is well-written and the information is clearly communicated. The data are analyzed appropriately, and the conclusions are justified.
However, several issues are noted which require consideration.
There is inconsistent or incorrect use of terminology used in connection with the field of implementation science. While the authors correctly note that implementation is a process, entailing a set prescribed activities, it is misleading to refer to an intervention as “well-implemented” or “high implementation” without a more formal analysis of the process that informs that implementation. Instead, the authors appear to be referencing the “reach” of the intervention, which, in this study, encompassed the proportion of participants who “adopted” the intervention. Thus, it is strongly suggested that the manuscript is carefully edited to more appropriately employ the terms implementation, adoption, and reach.
RESPONSE: Thank you for your positive words about the manuscript. We agree with your comment regarding implementation. We have changed the wording, i.e. in lines 50 and 54 it now says “intervention uptake” instead of implementation.
Line 44: “reasons for smoking initiation” – could be more clearly expressed to distinguish between individual reasons/motives and drivers of initiation, including risk factors such as low SEP. I suspect that the latter is more relevant.
RESPONSE: The lines 40 to 44 is about gender differences in smoking initiation. However, we agree that a little more about other reasons would fit in here. We therefore added a few lines about this.
Line 118: A reference is needed to support the OSC metric used in this paper
RESPONSE: We provided this reference, the new reference nr. 15.
Table 1 appears to have experienced formatting problems and is very hard to comprehend in its current form. The table is also confusing because multiple concepts are being communicated, including descriptive data and regression models. It is further confusing to see the overall proportion of girls vs. boys who signed a contract in the table, while underneath is the distribution of the sample by social class, rather than the actual proportion who signed the contract by social class (which appears in the figure). Revisions are needed to make the table more comprehensible, including the possibility of breaking the table into 2 smaller tables.
RESPONSE: We agree with this comment and have changed the table. We hope that you will also find it easier to read in the new version.
Lines 185-86: The authors claim that distribution of contracts was unbiased because they were systematically distributed through schools. However, this does not account for potential OSC variations between schools. The authors should provide more information about the socioeconomic positions of the schools that took part in the study, and explain whether a cluster analysis might have allowed better control of variations in OCS.
RESPONSE: Thanks for this comment. You are right about the variation between schools. However, variation between schools in Denmark are rather small, and would not as such be regarded as an issue here.
Round 2
Reviewer 1 Report
The authors responded my observations